# A Tuneable Pressure-Based Energy Harvester for Powering the Environmental Internet of Things

**DOI:** 10.3390/mi13111973

**Published:** 2022-11-14

**Authors:** Joshua Curry, Nick Harris, Neil White

**Affiliations:** Department of Electronics and Computer Science, University of Southampton, Southampton SO17 1BJ, UK

**Keywords:** energy harvesting, phase change materials, environmental sensing

## Abstract

As the internet of things expands to more remote locations, solutions are required for long-term remote powering of environmental sensing devices. In this publication, a device is presented which utilises the slow-moving diurnal temperature change present in many natural environments to produce electrical energy. This device utilises a novel actuator which harnesses temperature-dependent phase change to provide a variable force output, and this is combined with energy storage and release apparatus to convert the output force into electrical energy. Appropriate modelling is utilised to identify parameters for system tuning, and a final proof-of-concept solution is constructed and demonstrated to generate up to 10 mJ per 24 h period.

## 1. Introduction

The Internet Of Things (IoT) is expanding to more locations on our planet than ever before. Once far from internet connectivity, remote locations such as glaciers [1], fields [2] and underwater environments [3] are now becoming internet-connected and monitored by IoT sensors.

Historically, in these remote locations, systems have always been limited by connectivity. However with this constraint mitigated even in the most difficult regions by novel wireless sensor networks such as NB-IoT [4] and LoRaWAN [5], and new satellite connectivity for sensing devices such as swarm.space [6,7], the ethos of development shifts from systems which were once limited by connectivity to systems which are now limited by power.

Conventionally, solar photovoltaic technology has been by far the most well-used solution for powering remote sensing devices. However, as a technology, solar photovoltaic energy production is not without its drawbacks, especially in remote outdoor environments subject to constant change. Solar photovoltaic devices can be greatly influenced by changes in the environmental conditions during extended deployments such as the onset of dust and coverage by vegetation which can affect their output and limit the lifespan of remote sensing devices without maintenance.

Thus, there is a need for an energy harvesting system which can power these types of remote devices for sustained periods without these drawbacks, allowing for sensing devices to be deployed for much longer periods without requiring maintenance.

A potential energy source common to environments across the globe is diurnal temperature change. The daily movement of the sun over the environment and subsequent night causes temperatures in the day to consistently be higher than temperatures at night. This source of energy varies in magnitude across different latitudes, with deserts at the planet’s equator commonly seeing upwards of 30 ${}^{\circ }C$ temperature change in a 24-h period [8].

A novel energy harvesting solution has been identified which utilises phase-change materials to harvest energy from this diurnal temperature change. These materials, usually utilised in the form of a liquid in equilibrium with its vapour, typically exhibit a nonlinear pressure response to an increase in temperature. Established literature has identified a number of ways in which this energy can be extracted. Historically, however, most have relied on prefabricated components and present little potential for tuning [9].

In this paper, previous developments of a tuneable actuator [10] are harmonised with an energy extraction system to harness the energy available from diurnal temperature change into electrical energy. The system is constructed, tested and further tuneable factors identified to allow this type of energy harvester to be applied to a wide variety of outdoor environments across the globe.

## 2. Background

The utilisation of phase change materials to provide mechanical energy has been previously explored in modern engineering literature. The earliest application of this principle to a Commerical product began with the Atmos Clock, a self-winding clock which utilised a set of sprung bellows containing Chloroethane ($C_{2}H_{5}\mathrm{Cl}$) to power an escapement mechanism [11].

As the temperature of the clock rose and fell across a day with the diurnal temperature change present in its environment, the vapour pressure of chloroethane within the bellows changes in a similar fashion, producing mechanical force which can be utilised for the escapement. For a clock, this solution is intrinsically efficient, with two days of operation driven by as little as 1 ${}^{\circ }C$ temperature change [12].

Whilst powering mechanical devices is useful, researchers soon realised the potential of this type of device for use in powering electrical systems and sensing devices. The publication Energy harvesting from atmospheric variations by Ali et al. [12] explored the response of these chloroethane-filled bellows, which at the time were available inexpensively as a spare part for various modern iterations of the Atmos clock. Their work showed that temperature variations between 15 ${}^{\circ }C$ and 35 ${}^{\circ }C$ yielded substantial changes in the length of the sprung bellows, with displacements up to 35 mm which could be used to power subsequent energy conversion systems.

This publication was later followed up with a further endeavour into the field, with a theoretical exploration of an energy conversion device which could utilise this source of mechanical force variation to harvest electrical energy. The publication Conversion of atmospheric variations into electric power [13] proposes an energy conversion system based on substantial gearing and a clock spring mechanism which can be manually released to unwind mechanically stored energy into a small electric generator. Whilst extensively modelled and explored theoretically, this has yet to see an implemented system which enables the automatic harvesting of energy using mechanical force variation due to gas pressure response as a source.

In contrast to the direct gearing of force output, a publication by Zhao et al., Powering sensor nodes with ambient temperature changes [14] chose to combine Atmos clock bellows with a small sprung linear energy harvesters, providing multiple smaller pulses of output across the expansion and contraction of the Atmos bellows. This research yielded the first practical implementation of energy harvesting from this energy source, outputting 21 mJ of energy per cycle between 5 ${}^{\circ }C$ and 25 ${}^{\circ }C$.

Both of these systems utilise Chloroethane. With a boiling point of 278 K ( 12.3 ${}^{\circ }C$), Chloroethane has a well-matched temperature/pressure characteristic for use in temperate environments. However, is not the only chemical with a favourable pressure response to temperature change. In fact, any gas in equilibrium between its liquid phase and gaseous phase in sufficient concentration can provide a nonlinear response to temperature change, however in most cases always at a pressure higher than standard atmospheric pressure.

The first publication to explore the use of other gases for this type of mechanical energy harvester is Energy Harvesting across Temporal Gradients using Vaporisation by Xiao et al. [15]. This publication showed that a small quantity of butane, adequately constrained by a constant-force spring, could make use of changing environmental temperatures to move a model vehicle across a small distance.

In previous work, we initiated work into this use of other gases to drive a variable force output. A novel energy harvesting actuator for self-powered environmental sensors [10], defined an actuator which could be used to interface any gas with a favourable pressure response to a mechanical energy conversion system. Utilising this actuator, the gas mixture can be tuned to allow for favourable response, removing previous constraints on energy harvester designs to only use Chloroethane and obsolete spare parts with limited supply. It also presents a number of intrinsically tuneable factors, with resistance to gas expansion able to be tuned to a fine degree. The gas mixture can also be mixed with air to mute response and allow for higher temperature changes. In that publication, combinations of Propane and Butane were used due to their ease of availability and low toxicity.

In summary, the publications above highlight a number of key findings about the field. Firstly, usable energy has been generated from diurnal temperature change, in the order of 21 mJ per 20 ${}^{\circ }C$. Secondly, most current designs of energy harvester use a proprietary spare part from a clock which is no longer fabricated on a large scale. A novel actuator has been proposed which mitigates the various issues this poses and allows for the use of other gases which can be tuned for a higher pressure change across a desired temperature range. There is scope to integrate these two previous works to create a functional electrical energy harvester using a tuneable actuator to create a system which can functionally generate electrical energy in a wide variety of environments, and this publication expands on this.

## 3. System Design

The functional design of the proposed energy harvesting system consists of three main components. The Actuator houses the active materials which drive a pressure change from diurnal changes in temperature, and allows for the output to be extracted in the form of mechanical displacement. The converter utilises this in the form of torque, and allows for its storage until a point at which a release into an electric generator occurs at reasonable efficiency. At the point of release, the control electronics harvests the burst of electrical energy provided by the generator and offloads it into long term electrical storage. If enough energy is stored, the energy harvesting system-on-chip provides a regulated energy output to the mission. This mission could be a payload of a sensing device or similar electronic system which requires electrical energy to operate. As an example, a typical Bluetooth Low-Energy system requires approximately 100 μW [16] to operate.

Figure 1 shows a graphical representation of the architecture described above. The main challenge presented by this design is the large number of energy conversions which take place in the journey from pressure change to electrical potential energy. With each energy conversion there will be losses in the form of static and dynamic friction and electrical leakage. Subsystems need to be designed with overall system efficiency in mind, minimising losses which may through frictional forces also impact other parts of the system. This has been achieved by investigating a macro-sized system as a proof of concept.

### 3.1. Actuator Design

The design of the actuator is perhaps the most important element of the system. As the primary driver of mechanical energy from pressure change, it needs to be efficient and also allow for a large variety of tuneable factors, which can be altered to match the energy harvester to a specific environment. In order to design and specify this subsystem, first the pressure response of gases must be characterised.

Perhaps the most common method of simulating vapour pressure in use today is the Antoine Equation (Equation 1) [17], a semi-empirical relation which can be used to study the variation in vapour pressure across a range of temperatures.
(1)$$\log_{10}P=A-\frac{B}{C+T}.$$

A number of material-specific constants, *A*, *B* and *C* are used to give the Vapour Pressure *P* at a known temperature *T* (K). These constants are available for a wide variety of materials from established scientific sources such as the NIST Chemistry WebBook [18].
(2)$$p_{i}=p_{i}^{⋆}x_{i},$$

The responses of different gases can be combined into a single overall response by utilising Raoult’s law (Equation 2) [13]. Figure 2 shows the application of this law to calculate the vapour pressure of various mixtures of n-butane and propane.

In order to exhibit a vapour pressure, a volume must contain an adequate amount of gas such that an equilibrium is established between the gas in its liquid phase and its gaseous phase. The overall pressure response can be enhanced or muted by mixing in gases which have a higher pressure at the temperature of operation, or gas mixture such as air which has a much smaller pressure response to temperature change.

The Actuator design exhibits a number of critical design parameters which must be fulfilled in order for the device to operate. Firstly, the pressure of the gas mixture must never exceed the maximum pressure of the actuator assembly under normal operating conditions. In uncontrolled environments such as outdoors, this means that even in direct sunlight the actuator must be able to contain the gas pressure without critical failure. Secondly, the actuator must be able to extract the pressure of the gas as usable work, most easily attainable as a force output which travels through a set distance.

For these reasons, a double-acting pneumatic cylinder was chosen as the actuator, not least because most commercial options have a relatively high maximum pressure rating of 10 bar, which is easily high enough to accommodate mixtures of propane and butane at room temperature, but also because a number of extra tuneable factors can be specified through the cylinder selection. Though the use of a double-acting pneumatic cylinder, a resistance pressure can be applied to the gas through the second port of the cylinder which causes contraction of the piston rod at lower temperatures. At its simplest, this could be a volume of compressed air a number of times higher than the volume displaced by the cylinder at maximum extension, simulating a constant-force spring. The smaller pressure change in this volume throughout the travel of the cylinder, the smaller the temperature range throughout which the cylinder will extend.

A diagram of a double-acting pneumatic cylinder to be utilised in the actuator is shown in Figure 3.

The bore and piston rod of pneumatic cylinders are available in a wide variety of sizes, and this allows for tuning of the amount of force output by the pressure change of the gas. For example, as F=PA, where *F* is force output, *P* is pressure and *A* is area, a larger cylinder bore will give a significantly larger output force from the gas pressure. Similarly, a larger piston rod diameter will mean a smaller area on which the resistance pressure acts, meaning that the cylinder will experience a larger force output from the gas at a lower temperature and a higher resistance pressure will be required to compensate.

Pneumatic cylinders are also available with a wide variety of stroke lengths. As W=Fd where *W* is work done, *F* is force and d is distance travelled, a larger stroke length gives a larger energy output for a set overall force output by the cylinder. However, maximising this parameter will make integration of the cylinder with other apparatus difficult due to a much larger overall size of the system.

For these reasons, a *SMC CDG1BN20-100Z* pneumatic cylinder was chosen, with a bore of 20 mm, piston rod diameter of 8 mm and stroke of 100 mm.

In order to extract output energy from the actuator, a modulus-1 toothed gear rack was attached through a mechanical linkage to the end of the piston rod. This allowed for a spur gear to be placed in contact with the rack to take off output force as torque into the next subsystem.

### 3.2. Converter Design

In order to extract output energy from the actuator, a converter assembly is required to convert torque and rotation (force and distance) into electrical energy through work.

In its simplest form, the converter could be an electrical generator attached directly to the actuator output through direct gearing. The output rotation would occur slowly as the cylinder expands, and this could be geared up to directly drive an electric generator. However, in order to operate efficiently, electric generators usually need higher speeds of rotation, and as such an incredibly high gear ratio would be required. This would bring with it a variety of issues in the form of frictional losses, both dynamic friction due to the rotation of multiple moving parts, and static friction due to a higher number of rotational contact surfaces. As such, a method of energy storage and release is required to store up mechanical energy and release it at an efficient speed for a generator.

It is tempting to think that if the intrinsic link between temperature and pressure was negated, the actuator could simply be prevented from moving whilst pressures build, and then released when the output force exceeds a set value and allowed to expand, doing work through torque and rotation in the output spur gear. However, in this system this would not be possible because of the physical link between extension, temperature and pressure in the actuator. If allowed to rapidly expand, increasing the volume of the piston containing the gas substantially, the actuator would experience rapid cooling as the gas inside evaporates with the increase in volume. This would significantly affect the response, and as such an external mechanism of storage and release is required.

A torsional spring is proposed in [13] as a potential way to negate this problem, however in that publication it relied on a manual release of energy once the torque in the spring reached a set value. In order to design a functional energy harvester, this release point would have to be automatically controlled when the spring torque reached a certain value to allow autonomous operation.

Figure 4a shows the functional design of the converter assembly. In this design, a torsional spring is mounted in parallel to a source of work, in this case in parallel to the output torque of the actuator. One side of the torsional spring is connected to the input shaft, and one side is connected through gearing to an output shaft, and prevented from moving by a release arm. As the input shaft rotates, a release cam connected to it gradually approaches the release arm which is preventing the output side of the torsional spring from moving. When the torsional spring has been wound by a set rotation, the release cam pushes up the release arm, allowing the back of the spring to rapidly unwind and release the mechanical energy stored into the output shaft. As the input side of the shaft is held in place by the position of the actuator, all the stored energy unwinds through the back of the spring into the output.

This converter is tuneable in a variety of ways. Firstly, the starting position of the release cam can be set at time of fabrication to wind a set fraction of a rotation into the spring before release by bringing its starting position closer to, or further from the position of the release arm. The torsion in the spring can be set by utilising different materials internally, or by pre-winding the spring through a set amount and preventing it from unwinding fully. The size and gearing of the output shaft can also be altered to ensure a good match with any electrical energy generation system which is connected. In this design, we have focussed on the intrinsic operational design of this part to allow for a release of the sprung system at a specific input force and as such there is scope for optimisation of energy output.

Due to the large number of custom parts, 3D printing was selected as the method of fabrication of the vast majority of parts of the converter assembly, except for the torsional spring for which a clock spring was used. A 3D model of the parts used for the converter assembly can be found in Figure 4b.

In order to connect the actuator and converter together, a 20-tooth modulus-one spur gear was selected to transfer torque from the toothed rack attached to the actuator into the input shaft of the converter assembly. Due to the low number of teeth compared to the rack, multiple releases are possible as the actuator increases in length if the actuator provides enough force.

### 3.3. System Modelling

The response of the combined actuator-converter system can be predicted statically by considering the forces involved. As shown in Figure 3, the force output of the cylinder, negating frictional losses, can be shown as the sum of the two forces $F_{gas}$ and $F_{res}$.
(3)$$F_{out}=F_{gas}-F_{res}$$

These two forces can be represented using the pressures present inside the cylinder.
(4)$$F_{out}=P_{gas}A_{gas}-F_{res}A_{res}$$

If the mixture of gases in the cylinder consists of butane, propane and air which was present during filling, the pressure of the gas can be given by the following.
(5)$$P_{gas}=P_{butane}x_{butane}+P_{propane}x_{propane}+P_{air}x_{air}$$

The converter assembly can be simulated by a static force threshold, exerting $F_{mres}$ in the opposite direction to the actuator until exceeded, when release of energy occurs.

Utilising these equations in combination with (Equation 1), a simulation can be developed showing the temperature of release of a converter assembly with different resistance pressures and mole fractions of gas mixture versus air. For this simulation, a gas composition of 50% propane and 50% butane is assumed, as this is a commonly available pre-mix of gas.

Figure 5 shows the effect of varying resistance pressure on the release temperature of the converter assembly. As would be considered reasonable, a lower resistance pressure yields a lower release temperature. This shows great potential for an easily set-able tuning parameter, which is able to be varied to change the operation temperature to suit different environments.

Figure 6a shows the effect of varying proportion of gas mix versus air in the cylinder. As predicted, a lower proportion of gas results in the converter assembly releasing at a higher temperature, as the assembly needs to heat up much more before the release threshold is reached. Increasing the molar fraction of gas up to 100% (filled at vacuum) shows that the curve begins to flatten, meaning that a compromise could be found between a lower gas concentration in the cylinder (and therefore lower maximum pressure) and a reasonable release temperature that works well in the environment of operation.

This is confirmed by Figure 6b. We assume that a hard limit of 10 bar is set for the maximum allowable gas pressure in the actuator as the system would be subject to direct sunlight at times. A gas concentration of 100% would be unsuitable for this purpose as the maximum allowable pressure is breached at approximately 50 ${}^{\circ }C$, well within possibility in direct sunlight. By decreasing the concentration of gas to 60%, this maximum allowable temperature increases to above 70 ${}^{\circ }C$, and the release temperature can be tuned to the required value by using the resistance pressure.

## 4. Methodology

In order to verify the operation of the actuator and converter assembly, an experiment is designed to sweep the temperature of the apparatus across a range of temperatures similar to those found in temperate climates, with release temperatures at different resistance pressures compared. It was chosen to vary resistance pressure instead of gas concentration as it is much more controllable and measurable using a standard absolute pressure sensor.

In this experiment, the actuator is filled with Rothenberger Super100 [19], which our previous publication found could be modelled by a gas mixture of 50% Butane and 50% Propane. Due to experimental constraints, it was chosen to restrict the weight of gas within the system to below 3 g, such that an intrinsically safe air volume would be present at all times within the environmental chamber, even in the eventuality that all the gas content leaked from the actuator. Before filling, the actuator is chilled to −10 ${}^{\circ }C$ for an appropriate period such that the fill temperature of the gas and any air also present is known.

For this experiment, a resistance pressure range from 3 bar to 4 bar was selected, lying in the realm of positive temperatures shown in the simulation in Figure 5. A temperature range of −10 ${}^{\circ }C$ to 35 ${}^{\circ }C$ was chosen to ensure that the assembly reset after each operation, and that a release would occur for all resistance pressures in the range chosen.

As an example electrical converter, a standard stepper motor is connected to the converter output at a 1:6 gearing ratio, and one of the two phases is connected through a Schottky diode rectifier to a 6600 μF capacitor with a 1 kΩ bleed resistor to provide an indicator of when the converter released.

Figure 7a shows the input temperature profile repeatedly applied to the system over a number of hours. This temperature profile was chosen both to ensure that each iteration resulted in a release of energy from the converter, and that the system fully reset after each cycle. Alongside this, a slow temperature ramp through the region of release temperatures to be explored ensures that the temperature of the gas inside the actuator can always be approximated by the temperature of the outside of the actuator. Figure 7b shows the physical system in its experimental configuration in the environmental chamber.

In order to measure the temperature of the experimental apparatus, an Arduino microcontroller with Honeywell HIH-6120 temperature/humidity sensor is secured to the actuator and insulated with dense foam from the surrounding environment to give an approximate representation of gas temperature inside the actuator. The HIH-6120 temperature sensor has a quoted datasheet accuracy of ± 0.5 ${}^{\circ }C$.

To facilitate measurement of gas and resistance pressures, Honeywell PX3 absolute pressure sensors are attached to both ports of the dual-acting pneumatic cylinder and the ratiometric voltages output by these are monitored alongside the output capacitor voltage by a Siglent SDS1104X-E oscilloscope. The Honeywell PX3 pressure sensors have a datasheet accuracy of 0.25% FSS (Full Scale Span), which for a 100psi sensor equates to 0.017 bar.

## 5. Results and Discussion

In order to characterise the actuator-converter assembly to a high degree of accuracy, the temperature profile above was applied to the system at four different resistance pressures, with a minimum of 10 repeats for each pressure. Throughout the experiment, the system performed well, and an energy release was triggered during every cycle.

Figure 8 shows the sensor output from the pressure sensors connected to the system and the voltage on the output capacitor across a number of temperature sweeps. This graph indicates a number of key findings. Due to the larger volume of the resistance pressure vessel, the resistance pressure changes a very small amount across a temperature cycle, providing a more-or-less constant force resisting the expansion of the gas as intended. At each release of the converter system (indicated by peaks in the capacitor voltage), there is a significant drop in pressure seen in the gas inside the actuator. This is due to the fact that at a release point, the actuator is no longer restricted by the spring force of the converter and able to expand until the converter locks again. This expansion of the working gas inside causes it to cool significantly, reducing its pressure. There are two releases of the conversion system seen in each temperature cycle due to the gear ratio between the toothed rack of the actuator and spur gear of the converter. This shows another potential design factor which could be tuned to ensure multiple releases across an energy harvesting period. As can be seen in the figure, the two releases of the converter system occur at two different temperatures in the current design. This is due to the transient decrease in temperature inside the actuator as it is allowed to expand at the first release point adding a time factor for the actuator to re-equalise with the environmental temperature in the chamber. This effect would be much smaller in an environment with diurnal temperature change, as this experiment shows an accelerated response and across a longer time span, these two releases would occur at approximately the same temperature. For clarity, only the first release point will be considered and compared with appropriate modelling.

Figure 9 shows a plot of the temperature of the first release against a number of different resistance pressures, averaged from all repeats of the temperature cycle shown in Figure 8. This shows a clear positive correlation between release temperature and resistance pressure, proving that the system can be tuned to release energy as the temperature of the system surpasses different fixed temperatures.

In order to compare simulation results against experimental data, two key physical parameters of the system are required. The first of these, $F_{mres}$ can be found by finding the maximum pressure differential between $P_{gas}$ and $P_{res}$ before the converter assembly releases. Due to the high number of repeats conducted during this experiment, this value can be averaged to give a good approximation for the force required to actuate the converter.

The second variable which needs to be found is the mole fraction of the working gas versus air in the actuator. This can be found by comparing the pressure exhibited in the actuator at a set temperature to that of a static container filled with the gas as an equilibrium of liquid and vapour. By rearranging Equation (Equation 6) and assuming a volume of solely gas and air in the cylinder, $x_{air}$ can be substituted for $1-x_{gas}$, leading to Equation (Equation 7).
(6)$$P_{out}=P_{gas}x_{gas}-P_{res}x_{air}\;\mathrm{Assuming}\;x_{air}=\left(1-x_{gas}\right)$$
(7)$$x_{gas}=\frac{P_{out}-P_{res}}{P_{gas}-P_{res}}$$

Utilising data from a number of experimental repeats, an approximate gas mole fraction of $x_{gas}=0.7$ and a converter release force of $F_{mres}=40N$ can be found. These can then be utilised with the simulation methodology described above to yield a simulated release temperature across the same pressure range utilised in the experiment. The results of this simulation are also plotted in Figure 9, and a comparison of the trend-lines indicate that the proposed simulation defines an accurate representation of the system and is able to simulate the release temperature to within 1 ${}^{\circ }C$.

With a functional converter assembly which can release energy into an electric generator at a specifically tuned temperature, this experiment has proven that a device using phase-change materials as an energy source is viable. In this publication electrical energy output has not been optimised and merely used as an indicator of converter release. With one phase of the stepper motor generating a peak of 1.7 V in a 6600 μF capacitor, 10 mJ is generated by this device per release. This can be easily increased by also harvesting energy from the other phase of the stepper motor, left unconnected in this experiment. Alongside this, the speed of release can be tuned using gearing to spin the generator at a speed optimal for energy harvesting. Capacitors can also be introduced in series with the motor windings to cancel the inductive impedance of the motor, further increasing the energy output by a significant margin [20].

## 6. Conclusions

This paper describes a functional electrical energy harvester, which can make use of diurnal temperature change to power environmental sensing devices. Initial characterisation of design parameters has highlighted a number of parameters that can be optimised to enhance the performance of this type of energy harvester in a chosen environment. The physical design of the actuator can be varied, with larger bores and longer strokes allowing for larger output forces for a given energy converter. The temperature response of the actuator can be varied by changing gas mixtures, or its response muted by adding air to the mix. Gearing between the actuator and converter can be varied to tune the number of releases per temperature cycle experienced by the system, and between the converter and the electrical energy generator to ensure optimally efficient energy output. The force posed by, and release point of the converter assembly can also be adjusted to vary the amount of energy stored before a release into the generator. In summary, this paper presents an initial design framework from which such an energy harvester can be constructed. A resultant energy output of approximately 10 mJ per release has been shown without optimisation of the electrical energy harvesting component, indicating great potential for improvement. Armed with appropriate modelling, future endeavours will focus on the miniaturisation of this technology to implement a smaller centimetre-scale solution which could be built into IoT sensing devices. It is intended to showcase a functional IoT device for environmental monitoring which is capable of using this energy source to record and transmit sensor data in the near future.

## Figures and Tables

**Figure 1 micromachines-13-01973-f001:**
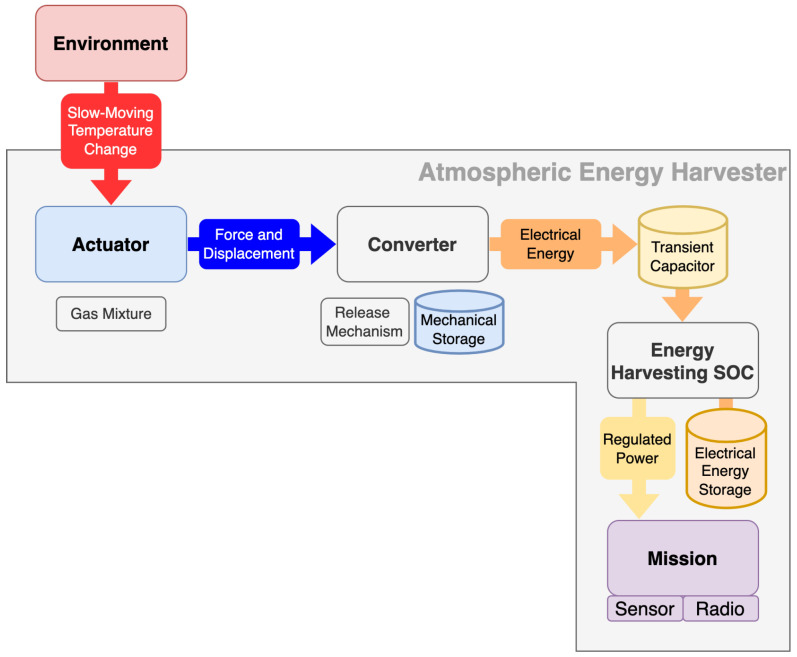
Block diagram of proposed energy harvesting system.

**Figure 2 micromachines-13-01973-f002:**
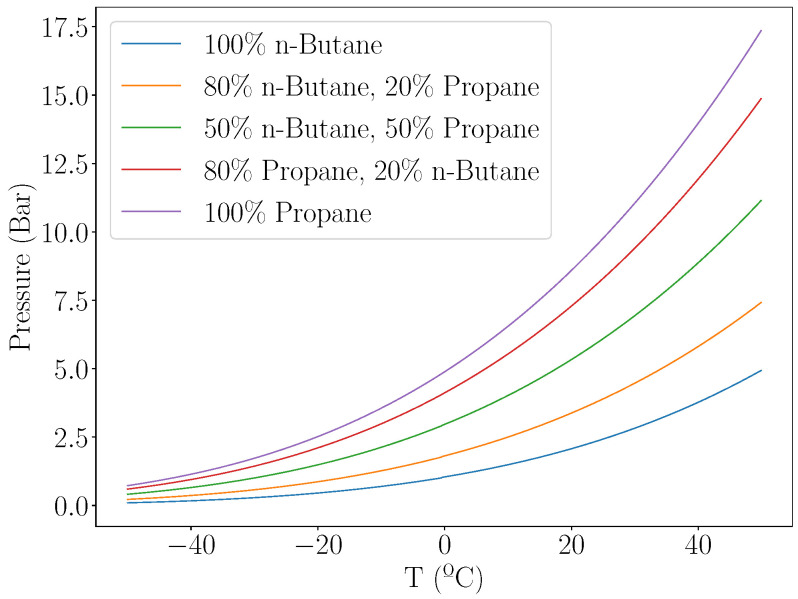
Calculation of vapour pressure of a range of mixtures of n-butane and propane.

**Figure 3 micromachines-13-01973-f003:**
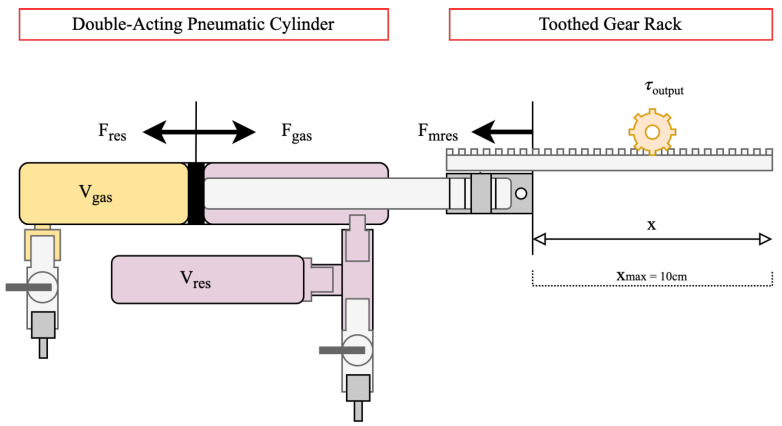
Functional diagram of double-acting pneumatic cylinder for use as energy-harvesting actuator.

**Figure 4 micromachines-13-01973-f004:**
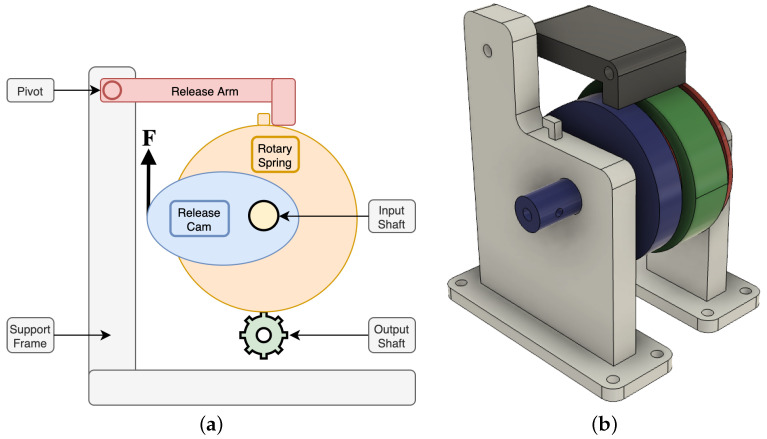
Design of converter system. (**a**) shows a functional diagram of the converter, (**b**) shows a 3D model of converter parts ready for fabrication.

**Figure 5 micromachines-13-01973-f005:**
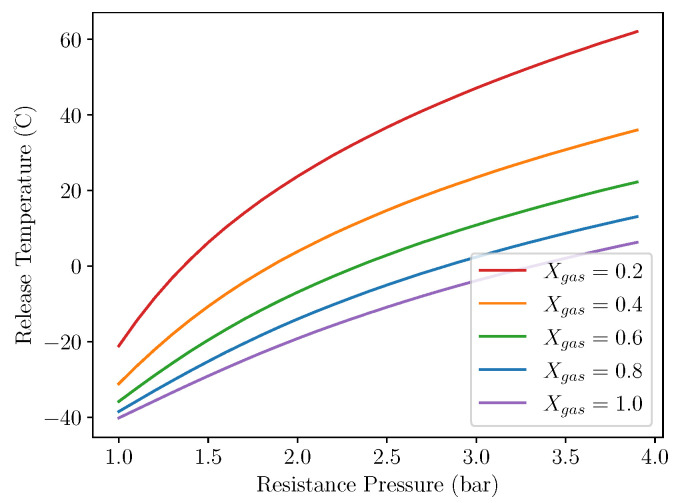
Release temperature vs resistance pressure for different gas fractions. $F_{mres}=10N$.

**Figure 6 micromachines-13-01973-f006:**
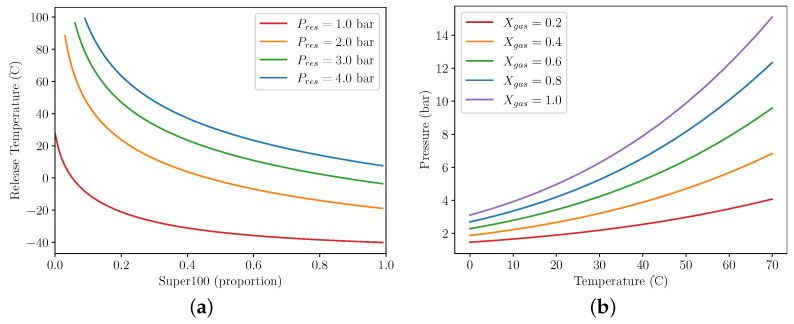
Release temperature and overall pressures for different gas fractions. (**a**) shows release temperature vs mole fraction for $F_{mres}=10N$. (**b**) shows overall gas pressure vs temperature.

**Figure 7 micromachines-13-01973-f007:**
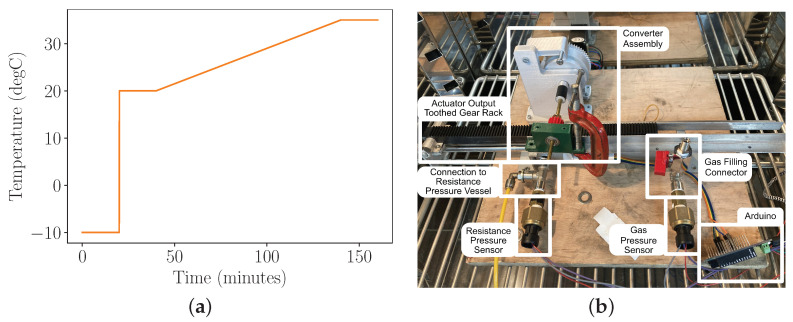
Experimental Parameters and Physical Setup. (**a**) shows the input temperature profile (duration 160 min, repeated). (**b**) shows layout of experiment in Environmental Chamber.

**Figure 8 micromachines-13-01973-f008:**
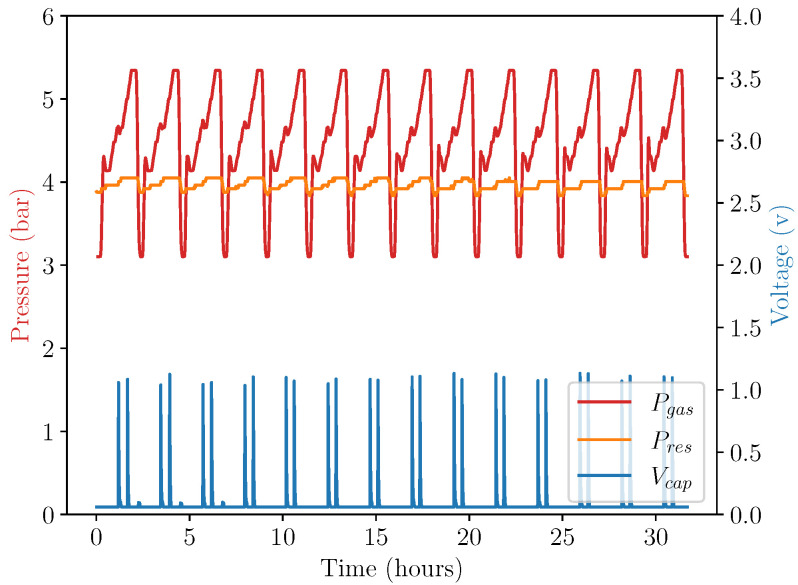
Actuator pressure and harvested voltage across multiple experimental repeats at $P_{res}=$1.4 bar. In this figure, the red line shows the pressure of the gas within the actuator, orange shows the resistance pressure (which changes very little due to its large volume), and the blue line shows the voltage on the output capacitor.

**Figure 9 micromachines-13-01973-f009:**
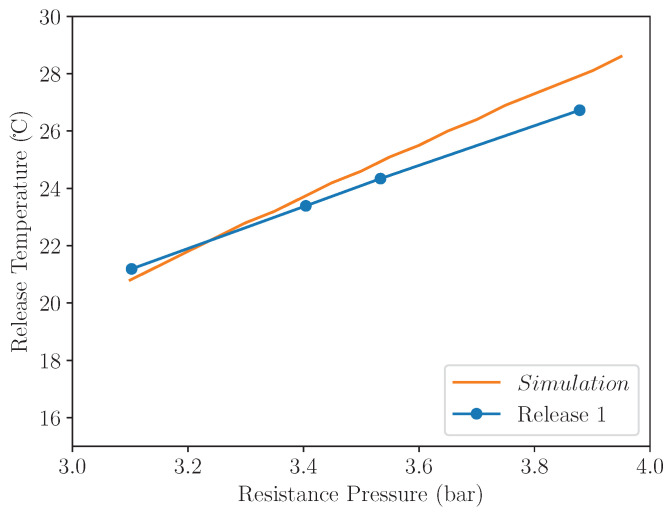
Release temperature across a range of resistance pressures, averaged across multiple experimental repeats. The blue line shows experimental results and the orange line shows the release temperature simulated by the system model across the same range of resistance pressures.

## Data Availability

The data presented in this study are available on request from the corresponding author.

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
