# Peer review of "A Tuneable Pressure-Based Energy Harvester for Powering the Environmental Internet of Things"

_micromachines, 2022, doi:10.3390/mi13111973_

Round 1

Reviewer 1 Report

The paper is an extension work of previously published article “a novel energy harvesting actuator for self-powered environmental sensors”. It utilizes a temperature-dependent phase change material to power remote sensing devices. The focus of this paper is to construct and analyze the energy harvester system. The article is well written , methodology is scientifically sound, results are clear and well presented. Thus, it is recommended to publish in Micromachines Journal with some proofreading.

Author Response

Many thanks for reviewing our manuscript. We have conducted a proofread to address any english language issues.

Reviewer 2 Report

This manuscript discusses about the atmospheric pressure/temperature based energy harvesting for environmental sensing devices. The work is well presented and easy to follow. I suggest considering the following comments while revising the manuscript.

The abstract is short and not having enough information about the article. It is suggested to describe briefly about the system and results so that the scope of the work is reflected in the abstract.

The accuracy of the devices used in the physical setup and the error for the various measurement needs to be presented.

In Fig. 7 (b), the size of the text can be increased for better visibility.

The abbreviation of "SOC" used in Fig. 1 can be included in Nomenclature.

In abstract it was mentioned as diurnal temperature change and in conclusion it was mentioned as diurnal pressure change. Both parameters are related but making it uniform like atmospheric pressure/temperature is looks good.  

Author Response

Many thanks for reviewing our manuscript.

The abstract has been extended to include a brief description of the system and results to give a better overall view of the scope of work addressed in the manuscript.

Datasheet values for errors in sensing apparatus have been added to section X to better characterise the experiment.

Fig. 7 (b) has ben adjusted for reasability.

SOC added to nomenclature.

Diurnal temperature/pressure change harmonised as Diurnal temperature change as it is the main factor of change.

A general proofread was also conducted to iron out any grammatical errors.